# The Strain Distribution Reconstructions Using GWO Algorithm and Verification by FBG Experimental Data

**Meng Zhang** [1,*], **Jingyan Wang** [1], **Xiao Xiong** [1], **Zihan Chen** [2], **Ying Gong** [1], **Sisi Gao** [1] **and Weifang Zhang** [3]

1   Space Environment and Reliability Division, Beijing Institute of Spacecraft System Engineering, 104 Youyi Rd., Haidian Dist., Beijing 100094, China

2   Institute of Remote Sensing Satellite, China Academy of Space Technology, 104 Youyi Rd., Haidian Dist., Beijing 100094, China

3   School of Reliability and Systems Engineering, Beihang University, 37 Xueyuan Rd., Haidian Dist., Beijing 100191, China

*   Correspondence: zhangmeng123@buaa.edu.com; Tel.: +86-135-816-29833

**Featured Application: The main purpose of this study is to present a strain profile demodulation method for a fiber Bragg grating (FBG) reflection spectrum based on the grey wolf optimizer (GWO) algorithm and a modified transfer-matrix method (TMM). A reconstructed non-uniform strain profile produced by the GWO algorithm was verified by experimental data obtained using the digital image correlation (DIC) method. The results showed that the target experimental strain profile was in good agreement with the reconstructed strain profile.**

**Abstract:** A structural strain reconstruction based on the grey wolf optimizer (GWO) algorithm using fiber Bragg grating (FBG) sensors is described in this paper. The fiber strain data obtained by the GWO algorithm and a modified transfer-matrix method (TMM) are verified by experimental data obtained using the digital image correlation (DIC) method. In the GWO algorithm, the optimization goal is set as the minimum error between the target experimental deformation spectrum and a random simulation strain spectrum, and the strain reconstruction is obtained by means of continuous iterative optimization. The validity of this method is confirmed by experimental strain data obtained by DIC, and the verification results show that the method proposed in this paper can be used as an accurate and efficient method of strain reconstruction.

**Keywords:** strain distribution reconstructions; GWO algorithm; fiber Bragg grating

## 1. Introduction

Cracking is a common form of damage in engineering applications, especially in aerospace structures. A recent report by the NASA Spacecraft Fault Management Workshop described small cracks found in the NZRYA module of the International Space Station (ISS). These fissures were described as superficial, but with the potential to spread over time. The authors of the report were unsure whether the cracks were responsible for any air leaks from the orbiting lab [1]. Such reports indicate the importance of detecting structural damage to spacecraft as early as possible, and this is now a matter of considerable interest to researchers. Fracture mechanics are now routinely applied in the early stages of structure design for the purpose of life prediction using damage tolerance design methods [2]. In addition, because structures only need to be repaired when a crack propagates to a certain dangerous length, conventional non-destructive detection techniques are unsuited for the requirements of long-term online monitoring. For this reason, it is necessary to detect crack propagation using structural health monitoring (SHM) techniques.

Fiber Bragg grating (FBG) sensors have exhibited great potential in structural health monitoring. In this regard, their light weight, high resolution, multiplexing capability, and immunity to electromagnetic fields can all be seen as advantageous. In SHM applications,

FBG sensors can monitor any damage or environmental impact during the operation process without compromising their structural strength, because shifts in central wavelength correspond to changes in strain/temperature. A uniform strain sensed along the axial direction of an FBG sensor leads to a Bragg wavelength shift, and this shift is linearly proportional to the constantly applied strain. In this way, FBG sensors can be used to identify the local strains upon a structure [3].

In spacecraft, cracks usually occur around holes in the mechanical structure, and a stress concentration zone involving stress singularity distribution and a strain gradient appears at the crack tip during the crack propagation process. Such an inhomogeneous crack-tip strain field phenomenon affects the response of FBGs. A deformation reflection spectrum response is indicated when the non-uniform axial force sensed by the FBGs varies along the grating direction [3]. However, a determination of structural strain distribution cannot be achieved by the Bragg wavelength shift algorithm alone when significant strain gradients appear around the FBGs [4]. Determining the best means of using a high-precision reconstruction algorithm to reconstruct axial non-uniform strain distribution from the reflection spectrum of FBGs can, therefore, be seen as an inverse problem, which many researchers have so far attempted to resolve [5]. The key to solving the inverse problem is to establish a database—including various simulation deformation spectra for sections of FBGs under different strains—by application of a modified transfer-matrix method (TMM). Previous studies have found good agreement between target reflection spectra profiles and reconstructed profiles obtained using modified TMM methods under strain field conditions [4]. A reconstructed strain may be considered as a simulation strain whose spectrum matches that of an experimental reflection with a high degree of accuracy and a minimal level of error tolerance. An optimization algorithm can then be adapted to deal with the problem of matching the simulation and experimental spectra, and can determine the strain with the minimum spectrum error. The reconstruction of the inhomogeneous strain field, or the fluctuations in stress–strain behavior, can be used to identify structural damage, such as cracks or impact damage [6].

A growing body of research suggests that intelligent heuristic algorithms perform well in solving this inverse reconstruction problem [7–11]. The authors of [7] demonstrated that a differential evolution algorithm was able to solve the inverse strain problem of FBGs. However, their algorithm was only verified by simulation data and lacked experimental data support. Swarm intelligence (SI) is another powerful form of computational intelligence (CI) which has been used to solve inverse optimization problems. The authors of [8] used a quantum-behaved Particle Swarm Optimization algorithm to reconstruct FBG sensor strain profiles. In another study, a dynamic Particle Swarm Optimization algorithm was used to reconstruct a non-uniform strain profile for the FBG sensor [9]. Finally, in [10], the reconstruction of non-uniform strain profiles by means of a genetic programming algorithm was described. Researchers have, therefore, confirmed the effectiveness of the optimization algorithm in solving the strain reconstruction problem. However, no experimental verification has yet been obtained with respect to the fatigue testing of aerospace structures. Researchers have also failed to consider the bonding layer effects of FBGs when glued onto such structures, or the cross-sensitivity between strain and temperature effects, so as to determine the best practice for engineering applications. There has also been little to no research into the verification of strain reconstruction algorithms by monitoring fatigue crack propagation in aluminum alloy structures using FBGs. The details of the contributions and limitations of previous research, the advantages of the proposed paper, and future applications are presented in Appendix A.

In this study, we use numerical spectral simulations in combination with experimental fatigue crack propagation monitoring using the digital image correlation (DIC) technique. Field strain analysis and damage assessment using DIC was originally developed using a two-dimensional correlation of deformed and undeformed digitized object images [12]. Subsequently, an ex situ DIC technique was developed with the ability to indicate the strain and size of the plastic zone at the crack tip during a single loading cycle. However, this

technique cannot provide information on crack tip strain over repeated fatigue loading cycles. In summary, we may say that the areas of interest covered by previous studies have been relatively small. However, clear theoretical analyses using the grey wolf optimizer (GWO) method have been applied in strain reconstruction to detect the actual strain profile of a real structure in a fatigue test. This method [13] has been tailored for a wide variety of optimization problems due to its advantages over other forms of swarm intelligence. GWO involves very few parameters, and no derived information is required in the initial search. In addition, it is simple, easy to use, flexible, scalable, and has a special capability to strike the right balance between exploration and exploitation during the search, which leads to favorable convergence. Consequently, in this study, we aimed to reconstruct full-field macroscopic strain measurements of the fatigue crack growth process in aluminum alloy structures using both FBGs and the GWO method.

This approach has four key advantages over the methods used in previous studies. First, the strain obtained by the digital image correlation (DIC) method is used to verify the reconstructed non-uniform strain reported by the FBG sensor. The accuracy of the DIC method has previously been confirmed [14]. Second, the difference in strain transfer caused by the bonding layer between the structure and the FBG sensor is taken into consideration. Third, the cross-sensitivity between the temperature and strain of FBG sensors in practical engineering applications is also taken into account by this method. Finally, and in contrast to traditional algorithmic methods, online performance analysis highlights the efficiency and practicality of GWO methods in solving the inverse problem of FBG. By such means, online analytical processing of strain data using FBGs can detect deformations and crack damage caused to spacecraft by debris impact and fatigue loading.

## 2. Reconstruction Algorithm

### 2.1. The GWO Algorithm

The grey wolf optimizer [15] is a metaheuristics algorithm which is one of many recently developed swarm intelligence methods. It has been tailored for a wide variety of optimization problems due to its advantages over other forms of swarm intelligence. GWO estimates global optimum in a similar way to other population-based algorithms, but its mathematical model is novel. It involves the relocation of one solution around another in n-dimensional search space. In doing so, it simulates the means by which grey wolves encircle their prey in nature. GWO has only one vector of position, and so requires less memory compared with Particle Swarm Optimization (PSO), which involves position and velocity vectors. In addition, GWO saves only three best solutions, while PSO saves a best solution for each particle, as well as the best solution so far obtained from all particles. The mathematical equations of PSO and GWO are also different.

GWO is a swarm intelligence technique which was first proposed in 2014 [16]. The inspiration for the GWO algorithm is the social intelligence exhibited by grey wolves in terms of their pack leadership and hunting behavior in the wild. In each pack of grey wolves, there is a shared social hierarchy that dictates power and domination (see Figure 1). The most powerful wolf is $\alpha$, and this animal leads the entire pack in hunting, migration and feeding. When the $\alpha$ wolf is absent through sickness or for some other reason, or when an $\alpha$ wolf dies, the strongest of the $\beta$ wolves takes over leadership of the pack [17]. Figure 1 shows how the combined power and domination of $\beta$ and $\delta$ are less than that of $\alpha$ and $\beta$. This kind of social intelligence lies at the heart of the GWO algorithm. It is inspired not only by the hierarchical behavior of grey wolves, but also their hunting approach. When hunting prey as a pack, grey wolves follow a set of efficient steps: chasing, encircling, harassing, and attacking. This allows them to hunt large animals as prey.

### 2.2. The Strain Reconstruction Algorithm

The authors of [14] found that a simulated spectrum based on an improved transfer-matrix algorithm exhibited a high degree of matching with a real-world experimental spectrum. A simulated spectrum obtained by such means can, therefore, be used for the

strain reconstruction of an FBG sensor. This can be achieved by iterating the error between the simulated spectrum and the target experimental spectrum under unknown strain distribution until the error requirements are matched. A GWO algorithm can then be used for parameters of global optimization of error between the experimental and simulated spectra. An arithmetic flow chart of the strain reconstruction algorithm of FBGs can be presented in Figure 2 as follows:

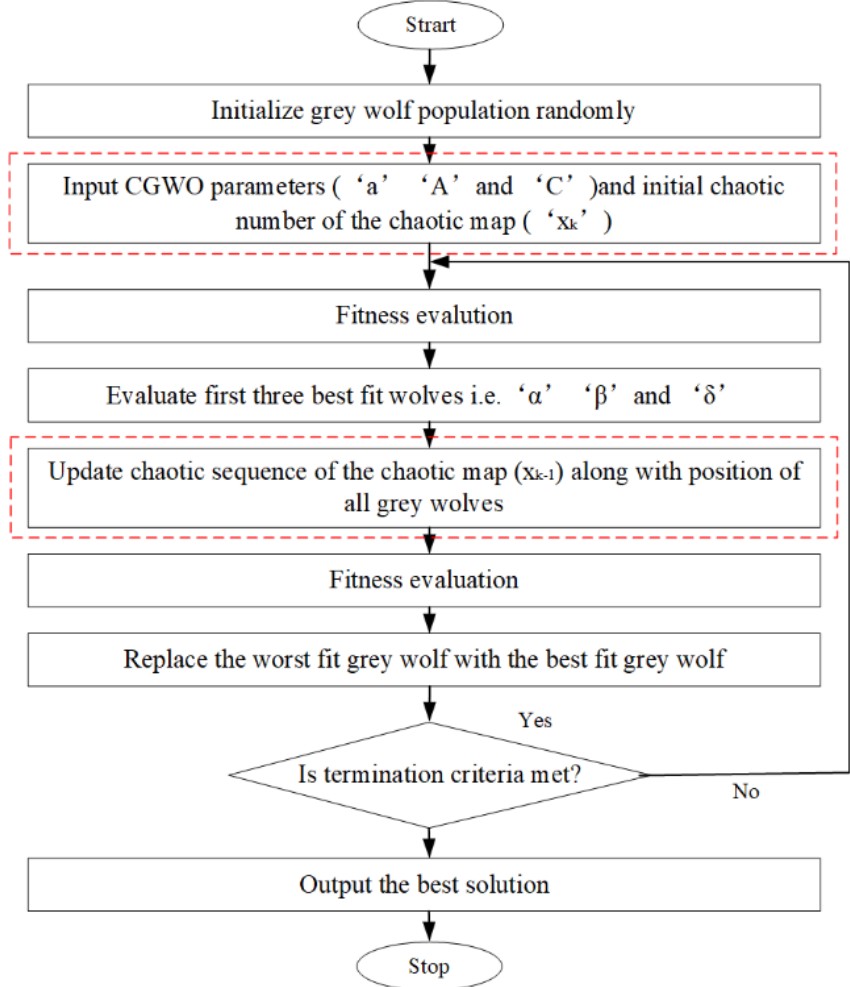

**Figure 1.** Flow chart of grey wolf optimizer algorithm.

This section briefly presents the mathematical model of the strain reconstruction of the FBG.

Step 1: The FBG simulated spectra database

When the modified TMM algorithm is used to simulate the reflection spectrum of FBG, the grating is divided into M-segments, and the strain distribution of each segment is presented approximately as uniform using the mean-field approximation method [19]. The inhomogeneous strain distribution sensed by the FBGs can then be transformed into the uniform strain distribution sensed by the M-segment of the FBGs. The reconstruction problem may also then be transformed, from a continuous and non-uniform strain distribution reconstruction problem into a discrete and uniform problem, greatly reducing the computational difficulty of the algorithm. The choice of M is limited by the spatial resolution used for the reconstruction strain profile. For example, if the reconstruction accuracy of the 10 mm FBG sensor is 0.1 mm, then the value of M is set as 100.

The key to the strain reconstruction method is the establishment of a simulated spectrum database of FBGs by the modified TMM method. The reconstruction strain distribution is calculated by comparing the simulated spectrum and experimental re-

flection spectra to obtain the minimum error. The simulation spectrum is calculated by the modified transfer matrix algorithm on the basis of a theoretical understanding, as follows: By dividing the non-uniform strain distribution $\varepsilon(z)$ sensed by the FBGs into enough discrete pieces, the M-segments can be presented as the uniform strain. The reconstructed non-uniform strain can then be expressed in terms of the uniform strain pieces of $\varepsilon^{(n)} = [\varepsilon_1^{(n)}, \varepsilon_2^{(n)}, \cdots, \varepsilon_M^{(n)}]$, $n = 1, \cdots, N$, and the Bragg wavelength segment of FBGs can be expressed as $\lambda_0 = [\lambda_0, \lambda_1, \cdots, \lambda_H]$.

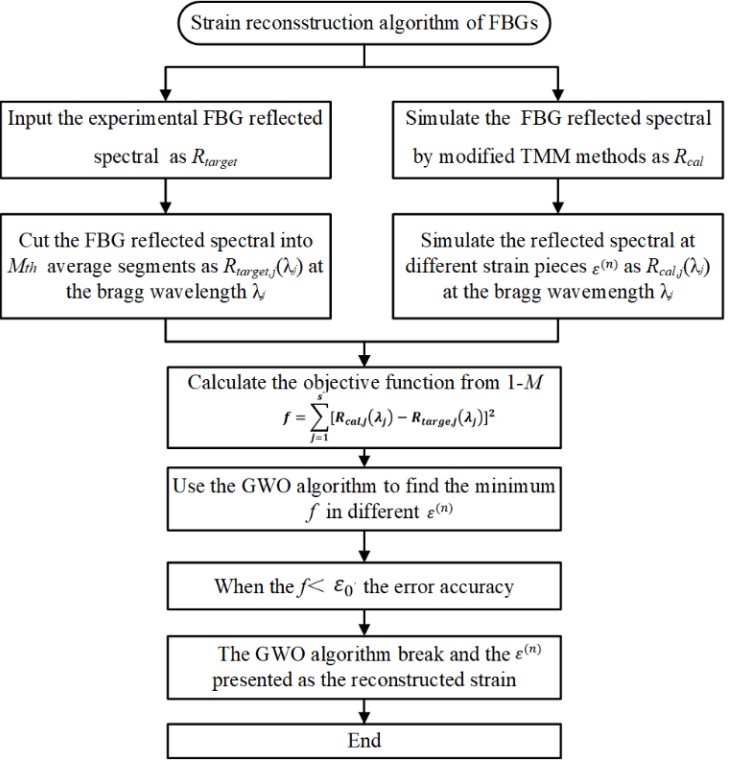

**Figure 2.** Flow chart of strain reconstruction algorithm of FBGs.

The modified transfer-matrix algorithm can be used to calculate the simulation spectrum of FBGs under assumed strain distribution. In the calculation function, $\varepsilon_m^{(n)}$ ($1 \le m \le M$) indicates the uniform strain value of the $M_{th}$ grating segment along the FBGs. Previous research [4] has indicated that the FBGs under non-uniform strain are mainly influenced by the strain gradient. Therefore, the first derivative $\varepsilon'(z)$ of the strain gradient should be used for the simulation spectrum calculation. However, the second derivative is too small and can be ignored for calculation purposes. The simulation reflection spectrum of FBGs at the Bragg wavelength of $r^{(n)} = r_1^{(n)}, r_2^{(n)}, \cdots, r_H^{(n)}$ can then be calculated using the modified transfer-matrix method.

Step 2: The strain reconstruction algorithm

As stated above, the non-uniform strain along FBG sensors can be expressed as a reconstruction problem that can be transformed into an optimization problem by minimizing the fitness sensitivity equations. The objective function $f = \sum_{j=1}^{s} \left[ R_{cal,j}(\lambda_j) - R_{targe,j}(\lambda_j) \right]^2$ can be calculated in terms of the simulation spectrum $R_{cal,j}(\lambda_j)$ and the real-world experimental spectrum $R_{targe,j}(\lambda_j)$, where $\lambda_j$ represents the $i_{th}$ Bragg wavelength and $S$ presents the total number of segments. When the appropriate optimization value $f$ is similar to the required error accuracy value $\varepsilon_0$, a global optimal solution is obtained. The grey wolf optimizer algorithm can, therefore, serve as a global method for the optimization problem that involves an optimum estimation for objective error function in terms of the difference between the simulation spectrum and the experimental reflection spectrum response. For each iteration, it automatically updates the search path at different regions in order to find

the minimum error of $\varepsilon_0$. The unknown strain distribution can then be obtained as the simulation strain distribution when determining the most optimal solution.

Step 3: Experimental data verification

Based on previously published research, and the specific optimization methods detailed above, a method of simulation verification can now be adopted [18]. In this study, the strain reconstruction algorithm obtained using the GWO method was verified by experimental strain data obtained by the DIC method. The simulation spectrum was calculated by the modified transfer-matrix algorithm based on the uniform strain of each grating segment, and the experimental spectrum was obtained using fiber Bragg grating sensor demodulation. Finally, the accuracy of the reconstructed strain of FBGs obtained by the algorithm was verified by the real experimental strain obtained by the DIC method.

## 3. Numerical Experimental Analysis

By means of a crack growth monitoring experiment based on fiber Bragg grating, the reflection spectrum of the FBG sensor and experimental strain information obtained using the DIC method were acquired. Then, based on the physical parameters of the FBG sensor, a reconstructed strain curve was obtained by the GWO algorithm combined with the TMM method. The reconstructed strain information was then compared with that of the experimental strain collected by the DIC method. The obtained strain profile was assumed to be real-world reference strain data in light of previous research, which proved the strain measurement accuracy of the DIC method [14].

### 3.1. Test Platform Based on DIC Method

An experimental platform for fatigue detection of hole-edge crack damage based on fiber Bragg grating sensors was constructed; this involved a fatigue crack measurement system, an optical sensing system, a DIC measurement system, and a fatigue load cycle system, as shown in Figure 3a. The specimens for the target system were made of 2024-T3 aluminum alloy, which is the alloy most widely used in aircraft structures. The Poisson's ratio of the materials was 0.33, the elastic modulus was 0.0731 Gpa, and the ultimate tensile strength and yield were 0.483 and 0.345 Gpa, respectively. A specimen with dimensions of $300 \times 100 \times 2$ mm and a 10 mm hole was placed in the plate center, into which a penetrating crack of 2 mm width had previously been introduced by electric discharge machining to promote the initiation of a fatigue crack. In the experiment, a uniform tensile load was applied to the bottom of the sub-specimen, and the top was bounded so that freedom was exhibited only in the z-axis direction. The frequency of the cyclic load was set to 10 Hz, the maximum load was 0.05 GPa, and the stress ratio was a sine wave signal of 0.1. In order to avoid the problem of delay caused by high stress, the load was set to 0.04 GPa during signal acquisition, as shown in Figure 3a.

In order to avoid abnormal experimental phenomena caused by operational errors, two parallel experimental procedures were used in this study. In the upper part of the test piece, two FBG sensors (FBG1 and FBG2) were bonded perpendicular to the crack propagation direction with epoxy resin glue of 1 mm thickness, with a fixed gluing length more than twice the length of the grating area, so that the best strain transfer efficiency could be obtained. FBG1 and FBG2 were also used as local sensors to detect the axial strain distribution of the patched area during crack growth, as shown in Figure 3b, and they were also used comparatively to determine the grating length's influence on the monitoring level. In order to eliminate the temperature influence on the wavelength shift of the FBG reflection spectrum, the FBG3 was bonded in a free-strain condition onto the back of the specimen that served as the control experiment with the FBG1. The central air conditioner of the experimental system was adjusted, and the temperature was set to 18.5 °C. The parameters of the FBG sensors are shown in Table 1.

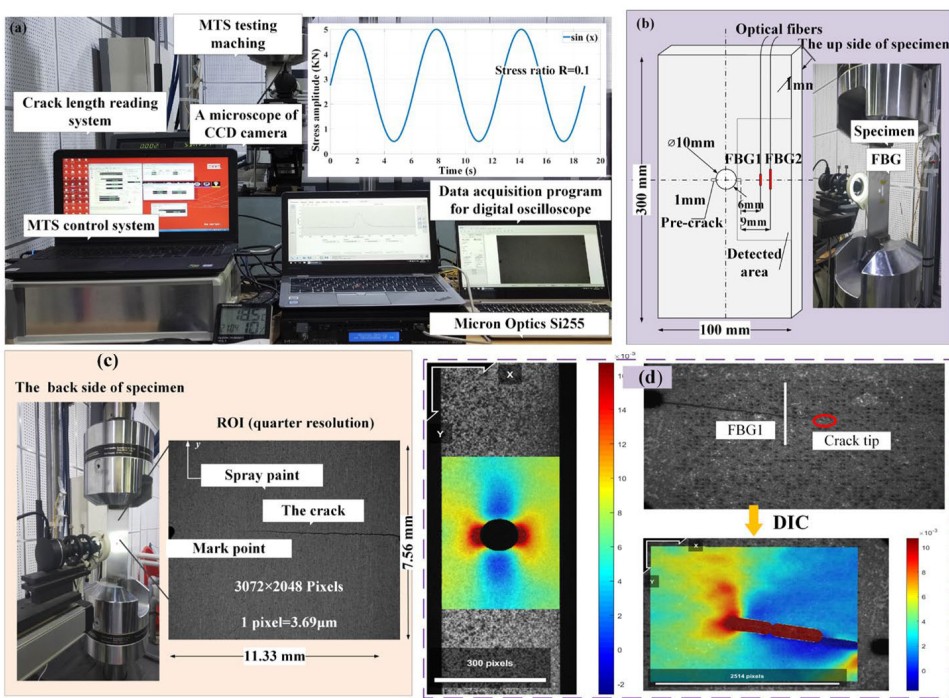

**Figure 3.** The FBG fatigue crack damage-monitoring test platform and the DIC processing method. (**a**) an experimental platform for fatigue detection of hole-edge crack damage; (**b**) the FBG sensor placement on the specimens; (**c**) an experimental platform for DIC measurement system; (**d**) the image processing techniques by the DIC method.

**Table 1.** The parameters of FBGs.

| Number of Sensors | FBG1 | FBG2 | FBG3 |
|---|---|---|---|
| Grating length (m) | 0.0051 | 0.0100 | 0.0051 |
| Effective refractive index | 1.450 | 1.458 | 1.450 |
| Bragg wavelength (nm) | 1550 | 1550 | 1550 |
| Poisson's ratio | 0.17 | 0.17 | 0.17 |
| Average index change | $2.0 \times 10^{-4}$ | $2.6 \times 10^{-4}$ | $2.0 \times 10^{-4}$ |
| Distance between FBG and original crack tip (mm) | 6 | 9 | 9 |
| Sensor side position | up | up | back |

Because the reference area in the DIC experiment was initially in a state of uniform strain, it was necessary to place FBG1 at a sufficient distance from the crack tip. In light of previous research findings, we set a distance of 6 mm between the center hole and FBG1, so that the radial distance between the two sensors was 3 mm. The 2 sensors (model FSSR5025) were used to monitor crack lengths of 1–12 mm, as shown in Figure 3c. For every 1 mm of crack extension, a low-light spectral demodulator device for FBG signal acquisition was used to obtain the reflection spectrum deformation. The wavelength test step size was 0.01 nm, which was the same as the optical fiber grating reflected spectra accuracy obtained by the TMM simulation. On the back side of the specimen, a plate was used to design the DIC experiment (specimen 1), as shown in Figure 3d.

To investigate strain distribution and gradient close to a growing fatigue crack tip, experimental DIC measurements were used to obtain strain fields over whole-plate areas, as well as high-resolution strain fields for differing crack lengths under differing fatigue loading cycles. To identify alterations in different observation areas, the marks were labeled to determine the strain field within a region of interest (ROI) for the specimen undergoing deformation by crack propagation. In DIC, image processing techniques are employed to obtain a one-to-one correspondence between subsets in the initial unformed picture and in the subsequent deformed images. In this study, strain information for the reference

configuration was obtained through location transformation of the matching subsets. Additionally, a spacing parameter was applied to reduce computational cost. Finally, a grid was produced which contained strain information with respect to the reference configuration. This can be referred to as a Lagrangian strain. The strain fields were then either reduced or interpolated to form a "continuous" strain field, as shown in Figure 3d. The strain curves of the FBGs can be seen in Figure 3e.

### 3.2. FBG Spectrum Reconstruction and DIC Method Verification

In this section, the reflection spectra of FBGs in an experimental test state are modeled and analyzed. The simulation reflection spectra of the FBGs obtained by the modified transfer-matrix method, as well as the bonded strain transfer coefficient (the strain transmissibility from the base aluminum alloy substrate to the fiber core) are then compared with the experimental reflected spectra. In order to test the accuracy of the FBG reflection spectrum simulation modeling method, the strain obtained by DIC simulation is used for the method validation data, as shown in Figure 4.

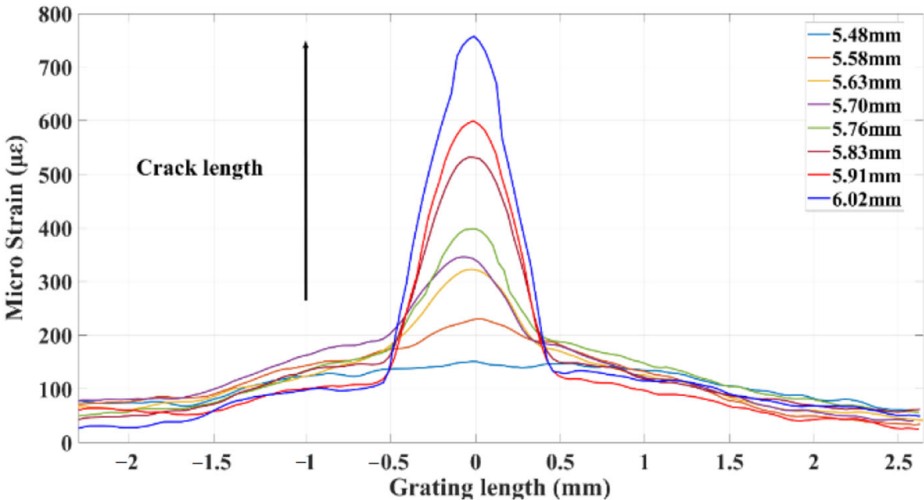

**Figure 4.** Non-uniform strain distribution at sensor positions of FBG1 obtained by the DIC method.

In line with previous research, the strain transfer efficiency was considered as a = 0.9 when fibers were bonded using epoxy resin adhesive. The influence of strain transfer between the glue and the sensors was incorporated into the modified strain distribution. The modified strain was equal to the product of the strain transfer efficiency and the strain extracted by the DIC method. The corrected sensor strain information was then substituted into the improved transfer-matrix algorithm as an input in order to obtain a simulation spectrum of higher accuracy. Finally, the simulated spectrum (as shown in Figure 5) was compared with the experimental spectrum (as shown in Figure 6) to verify the accuracy of the strain reconstruction.

By comparing the reflection spectrum obtained by the simulation method (Figure 5) with the reflection spectrum obtained by the actual experiment (Figure 6), it can be clearly seen that the aforementioned simulation method based on the strain transfer coefficient and improved transfer-matrix method using FBG physical parameters can effectively simulate the experimental FBG reflection spectrum curve. In light of this finding, a verification of the experimental FBG data in terms of the strain reconstruction method based on the GWO algorithm can be carried out. This method of verification provides a basis for the next step in reconstructing the experimental strain distribution of FBGs.

### 3.3. Strain Reconstruction Method Using GWO Algorithm

The previous section described how the research based on the strain transfer of the bonded FBGs and the spectral simulation algorithm based on the modified transmission matrix algorithm was validated. We can now describe how the strain reconstruction of the



reflection spectrum response of the test FBGs was carried out using the GWO optimization algorithm, when the number of wolves was set to 15. At the end of the optimization iteration, the mean squared error of both the simulation and experimental spectra was less than 1 dB$^2$. The target error value set here was larger than that used in the simulation process because the signals obtained in an actual experiment always contain environmental noise as a result of photoelectric conversion processing. For this reason, it was necessary to include certain noise-redundancy information in the experimental spectral reconstruction.

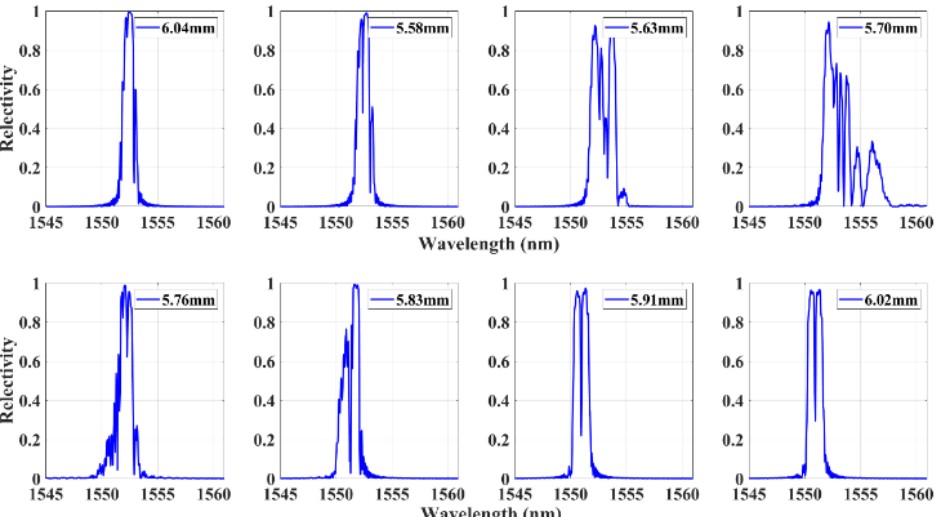

**Figure 5.** Simulation reflection spectra of FBG1 based on the modified strain distribution for different crack lengths.

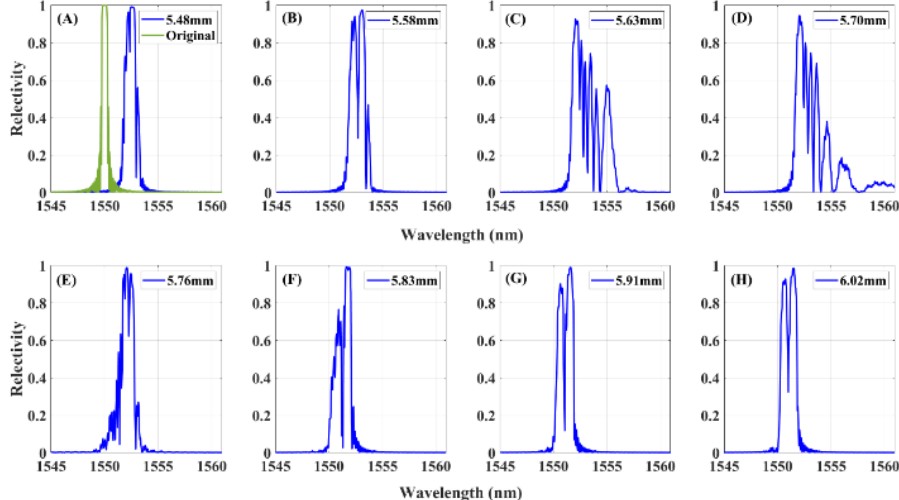

**Figure 6.** Experimental reflection spectra of FBG1 sensor for different crack lengths. (**A**) 5.48 mm; (**B**) 5.58 mm; (**C**) 5.63 mm; (**D**) 5.70 mm; (**E**) 5.76 mm; (**F**) 5.83 mm; (**G**) 5.91 mm; (**H**) 6.02 mm.

Figure 7 shows that the non-uniform strain distribution curve reconstructed by the GWO algorithm exhibits a high level of coincidence with the strain distribution curve collected in the experimental test. The effectivity and superiority of the algorithms are, therefore, demonstrated by the FBG experiment data. Moreover, the reconstructed strain distribution also takes the effects of strain transfer into account.

Finally, the precision of strain distribution reconstructions using the GWO algorithm under different crack lengths, in terms of RMSEs between the experimental and reconstruction strain values, can also be demonstrated, as shown in Table 2.

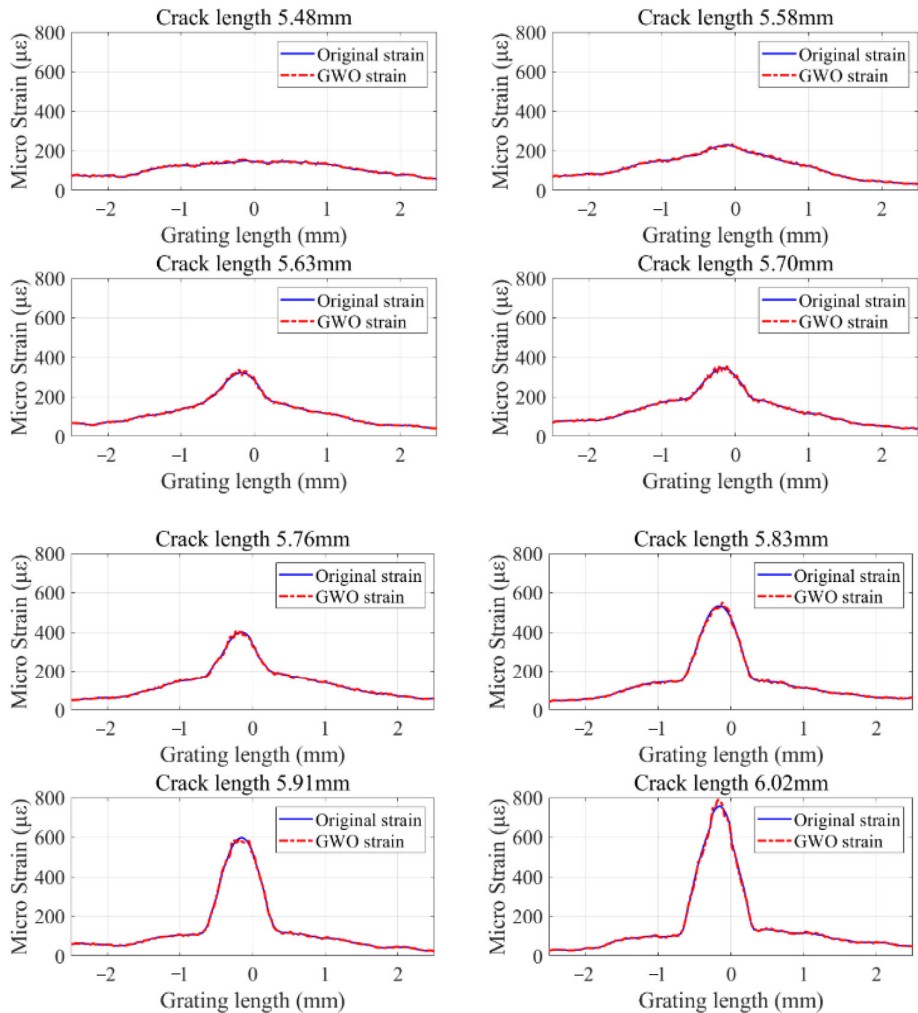

**Figure 7.** Calculations of non-uniform strains of FBG1 based on the DIC method.

**Table 2.** Strain reconstruction accuracy of experimental FBG reflection spectra by GWO algorithm.

| Crack length (mm) | 5.48 | 5.58 | 5.63 | 5.70 |
|---|---|---|---|---|
| RMSE (dB$^2$) | 0.1434 | 0.1511 | 0.1528 | 0.1441 |
| Crack length (mm) | 5.76 | 5.83 | 5.91 | 6.02 |
| RMSE (dB$^2$) | 0.1578 | 0.1502 | 0.1653 | 0.1666 |

*3.4. Comparison of Different Strain Reconstruction Algorithms with Experimental Data*

To compare the algorithm proposed here with other, more traditional algorithms used in previous studies, we also used RMSE as a quantitative evaluation index [6,19]. The RMSE results for experimental and reconstruction strains produced by different swarm intelligence techniques, i.e., the differential evolution algorithm, the quantum-behaved Particle Swarm Optimization algorithm, and the dynamic Particle Swarm Optimization algorithm, can be seen in Figure 8. These RMSE results are also related to different crack lengths during the damage monitoring experience, to further test the stability and accuracy of the reconstruction algorithm. Upon comparison with the traditional algorithms previously mentioned, the proposed GWO method presented the best strain reconstruction performance with the lowest RMSE. In short, the inverse problem in a real fatigue loading test can be effectively solved by use of the GWO method in a strain reconstruction algorithm.

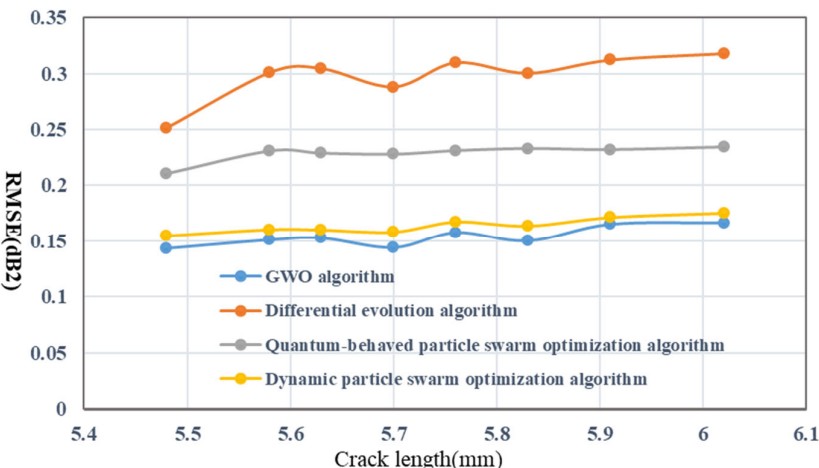

**Figure 8.** RMSEs of four strain reconstruction algorithms for different crack lengths.

## 4. Conclusions

To monitor large-scale deformation and crack damage to the structures of spacecraft, a non-uniform strain distribution of the FBGs affected by the sharp strain gradient should be reconstructed. In this paper, the GWO method combined with the modified transfer-matrix method is proposed to deal with the inverse problem. A theoretical analysis of the strain reconstruction algorithm was carried out based on a simulation spectral database obtained by the modified TMM method and a GWO optimization algorithm. The GWO method was found to produce the best simulation spectra, exhibiting high matching accuracy with the experimental spectra. The strain under the best-matching simulation spectrum could then be assumed to be the real-world experimental strain. In addition, various experimental spectra—on the basis of the crack monitoring fatigue loading platform—were used to verify the algorithm's performance in real-world application. The local strain field at the crack tip of an individual specimen was also measured, using non-contact DIC technology, during our experiments. Taken together, our experimental findings suggest a new and very useful way to identify the reconstructed strain distribution of FBGs by GWO and modified TMM methods, and our methods produced better RMSE results than those of other, more traditional algorithms. Based on the above, we suggest that the algorithm proposed herein may be used with actual in-orbit telemetry-obtained data or on-orbit real-time data in the future.

**Author Contributions:** M.Z. conceived the key idea and designed the experiments, J.W. provided the academic support and checked the manuscript, and X.X. and Z.C. performed the experiment. Data curation, X.X.; formal analysis, Z.C., S.G. and W.Z.; methodology, M.Z.; validation, J.W. and Y.G. All authors have read and agreed to the published version of the manuscript.

**Funding:** This research received no external funding.

**Institutional Review Board Statement:** Not applicable.

**Informed Consent Statement:** Not applicable.

**Data Availability Statement:** The data has been provided in this paper, and no additional data availability.

**Conflicts of Interest:** The authors declare no conflict of interest.

## Appendix A

The contributions and limitations of the previous research in different periods are discussed here in Table A1. Moreover, future applications are discussed in the following table, Table A2.

**Table A1.** The comparative table of our team's contribution with the previous research.

| Previous | The Contribution | The Limitation |
|---|---|---|
| Reconstruction of strain distribution in fiber Bragg gratings with differential evolution algorithm. | 1. This paper proposes the inverse problem of strain reconstruction of FBGs for the first time. 2. The differential evolution algorithm is used to solve the inverse problem. 3. The proposed linear and non-linear strain profiles are reconstructed based on the reflection spectra to verify the performance of the strain reconstruction algorithm. | The above two methods illustrate that the inverse problem of structural strain reconstruction can be solved through an optimization algorithm. The following are the unresolved issues in the above two papers: 1. In this paper, the traditional transfer matrix algorithm is adopted instead of the modified transfer matrix algorithm, resulting in relatively low accuracy of the simulation spectrum. According to the published articles, the modified transfer matrix algorithm shows better performance than the traditional transfer matrix algorithm in the distortion of spectral simulation caused by the strain gradient. |
| Quantum-behaved particle swarm optimization algorithm for the reconstruction of fiber Bragg grating sensor strain profiles. | 1. A quantum-behaved particle swarm optimization algorithm is proposed, which calculates the reflection spectrum of the FBG by combining the transfer matrix method and demodulates the strain profile along the FBG from the reflection spectrum. 2. The validity of the method is verified through numerical example reconstructions of FBG sensor simulated strain profile cases. Numerical simulations reveal that there is good agreement between the original strain profile and the reconstructed strain profile. | 2. Without considering the experimental strain, linear and quadratic simulation strain distributions are used to validate the algorithm. 3. According to the paper published by our research team, when the FBG sensors are used as strain sensors for structural health monitoring, the sensed strain is not completely assumed to be a linear distribution or a quadratic distribution. Only when the FBGs are far away from the damage; the sensed strain distribution approximates to a linear distribution. Then, when the crack damage grows close to the FBGs, and the distance between the crack tip and the FBGs is very small, the strain distribution sensed by the FBGs is assumed to be quadratic. Subsequently, when the structural crack damage passes through the FBG sensor, the strain distribution sensed by the FBGs is cubic. |
| Reconstruction of the non-uniform strain profile for fiber Bragg grating using dynamic particle swarm optimization algorithm and its experimental verification | 1. Compared with the previous two papers, the structural pasting experiment of the FBG sensor is firstly proposed in this paper. However, this paper does not consider fatigue loading, but rather, static loading. 2. In this paper, the traditional transfer matrix method is replaced by the modified transfer matrix method. 3. The DPSO algorithm is presented to demodulate the strain profile along the reflection spectrum of the FBG. 4. The feasibility and validity of the proposed algorithm are verified by simulation examples and experiment results. | 1. The experiment in this paper is still under static loading. Actual strain from structural damage characteristics, including impact damage and crack damage, are not simulated. Moreover, the experimental strain data mentioned in this paper are not the real strain data sensed by the FBG sensor, but the simulated strain data obtained through structural finite element simulation. 2. It is worth noting that the algorithm in this paper is not compared with other algorithms. 3. The FBG sensor is always affected by strain and temperature. Correspondingly, the spectral deformation of the FBG reflection may be caused by changes in temperature or strain. Furthermore, the cross-sensitivity of strain and temperature was not considered in the previous article. Therefore, it is inappropriate to use the reflected spectrum to reconstruct the strain without considering the temperature effect in the experiment. |

**Table A1.** *Cont.*

| Previous | The Contribution | The Limitation |
|---|---|---|
| Research on non-uniform strain profile reconstruction along fiber Bragg grating via genetic programming algorithm and interrelated experimental verification | 1. The experiment proposed in this paper is consistent with Wang's experiment design. Among them, the difference lies in the use of different optimization algorithms. 2. Based on the modified transfer matrix and genetic programming algorithm, a new heuristic strategy for reconstruction of nonuniform strain profile along FBGs is proposed. | 4. The FBG sensor is usually glued onto metal structures for sensor monitoring. The existence of the adhesive layer causes the strain transmission rate from the structural substrate to the fiber core not to be 1. In this case, it is necessary to consider the gluing effect of the sensor. **In our proposed paper:** The actual strain sensed by the FBG sensor in the crack damage monitoring experiment is obtained by the DIC method. In order to verify the effectiveness of the algorithm in this paper, three algorithms (DPSO, quantum-behaved particle swarm optimization algorithm, differential evolution algorithm) are used as comparison algorithms to compare with the GWO method proposed in this paper. The results show that the algorithm proposed in this paper is the best.The experimental design takes into account the cross-sensitivity of strain and temperature. Based on the relevant data in our published research patents, the strain reconstruction algorithm in this paper is chosen to have a strain transfer efficiency of 0.9. |
| Non-uniform strain field reconstruction of FBG using an adaptive Nelder–Mead algorithm. | 1. Differently from previous studies, this paper designed a complete specimen loading and spectrum collection system. Based on the previous experimental analysis, the strain data obtained by finite element simulation is still used as the real test data for the comparison of strain reconstruction. 2. This paper presents an adaptive Nelder–Mead algorithm, which inversely reconstructs the non-uniform strain field using the reflected intensity spectrum of the FBG sensor. | |

**Table A2.** Future applications.

| Future application | Regarding engineering application for space vehicle structures, especially spacecraft and platforms operating in orbit for a long time, it is necessary to monitor the structural damage considering the impact of space debris and the strong impact of launch into orbit. At present, the sensors used for the monitoring of space structures mainly include strain sensors, optical fiber sensors, and piezoelectric sensors. Among them, the strain sensor is a heavy point sensor with complex wiring which has been gradually replaced by the FBG sensors. Since the International Space Station (ISS) knot crack damage event in 2021, health monitoring based on FBG sensors has been gradually introduced to deal with the threat of damage to spacecraft caused by space debris. For example, the optical fiber demodulation equipment was carried into radiation and strain monitoring by the EU in 2021. Currently, FBG sensors, as structural strain sensors, mainly monitor the average regional strain of the structure. When the structure is slightly damaged, there may not be a large strain gradient jump. Thus, more attention should be paid to the strain information brought by the deformation spectrum. Through the combination of the strain reconstruction method based on the FBG sensor proposed in this paper and the structural finite element model, the real-time strain nephogram of the entire structure can be obtained. Based on the obtained strain cloud map information, the structural health status can be monitored on orbit. |
|---|---|

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
