# Peer review of "The Strain Distribution Reconstructions Using GWO Algorithm and Verification by FBG Experimental Data"

_applsci, doi:10.3390/app13031259_

Round 1

Reviewer 1 Report

(1)the garmmar throughout the paper should be checked and improved. For example, in the first sentence of the introduction, uniformly could not be used this way. and "in this way" should be "In this way".

(2) the state-of -the-art of the research in the introduction should be further explained.

(3) the method in section 2 is too simple, the main process should be introduced in detail.

(4) although the experimental data is proposed, does other method can reconstruct the strain? What is the differene? 

(5) the novelty of this paper should be further clarified.

Author Response

Thank you very much for your useful comments and suggestions on the content and language of our manuscript. Your comments are very valuable and helpful for revising and improving our work. We have seriously considered each comment and modified the manuscript accordingly. The detail response of the suggestions all has been presented in the attachment.

Reviewer 2 Report

Dear Authors,

The paper doesn't use any innovative techniques.

Good luck.

Author Response

(The authors gave the same response as above.)

Reviewer 3 Report

The article is not well prepared in general terms, and it is not sufficient in terms of organization. Therefore, the study cannot be accepted as it is. Some of my comments are summarized below.

1) There are quite a few mistakes in the article. Check out the grammar.

2) Introduction section is not well prepared. It could be expanded.

3) It should be clearly stated what the problem is.

4) Also, your contribution to the literature should be clearly stated.

5) Explanations and full names of some abbreviations should be stated in the first place in the text. Keep the abstract separate.

6) What are the studies in the literature? It should be added, and their pros and cons should be discussed

7) The proposed method is definitely not well explained and has been poorly prepared. It should be supported with pseudo-code and equations along with enough explanation.

8) This study seems to lack innovation. What exactly was the innovation considered?

9) It should be clearly explained how the GWO algorithm is used

10) Why did you use the GWO algorithm?

11) Comparisons with other methods (at least with other meta-heuristic algorithms) should be made.

12) The conclusion section should be rewritten extensively.

13) It is recommended to include future studies.

Author Response

(The authors gave the same response as above.)

Reviewer 4 Report

The authors present an algorithm to identify and track the non-uniform strain reconstruction of spacecraft metal structure using multiple FBGs. The validity of this method was confirmed by the experimental strain data obtained by the DIC method, and the verification results show that the method proposed in this paper can be used to realize the accurate and efficient strain reconstruction.

I would recommend this work be published once the following points are highlighted:

1)  Page 1 – this paragraph needs a bit more description at its beginning, such to the contribution and highlight comparing to the previous research. And the Comment should be made on this broader application for the developed algorithm.

2) Page 2line147. What was an influence of the gluing area? What are the works of the above three FBG sensors at present?

 3) It should be noted that in addition to fibre Bragg Gratings this work is also very relevant to the cross-sensitivity between temperature and strain? Here the authors should discus about the instabilities generated by the temperature effects and made a cross-sensitivity analysis.

4) The manuscript should give more material parameters information of the metal plates in the experiment, and the Young’s modulus and the tensile load is usually given in GPa not of Mpa – change unit magnitude to GPa.

5)       The authors should check if the citation in reference with the current formatting.

Author Response

Thank you very much for your useful comments and suggestions on the content and language of our manuscript. Your comments are very valuable and helpful for revising and improving our work. We have seriously considered each comment and modified the manuscript accordingly. The detailed corrections are presented in the attachment.

Round 2

Reviewer 1 Report

The revised version well reflects my comments, I think it is ready for publication.

Author Response

Thank you very much for your useful comments and suggestions on the language of our manuscript. The English grammar part has also been modified by MDPI English editing service, including grammar checking, spelling, punctuation and some improvements of style.

Reviewer 2 Report

Dear authors,

You have improved the quality of the article with the structural changes.

Regards.

Author Response

(The authors gave the same response as above.)

Reviewer 3 Report

When compared with the first version of the article, serious changes and adjustments were made. In this state, the article has reached an acceptable level and responses have been written carefully on my comments. However, it is accepted provided that the following corrections are made.

1) Table 1, which you have added to the response file, should be attached to the article as appropriate. If necessary, it can be added to the article as more than one table.

2) There seems to be a typo in the reference on line 306. It should be corrected.

3) In some abbreviations, you missed the upper and lower case characters. get it fixed. For example fiber Bragg grating (FBG), grey wolf optimizer (GWO), Particle Swarm Optimization (PSO), etc.

4) A clearer and more explanatory comment was expected for the 10th question. The more persuasive and simple you write, the more useful it will be for the readers. So, highlight this in the article.

5 Why did he use reference number 17? Removable.

Author Response

(The authors gave the same response as above.)
